# Reward-rational (implicit) choice:
# A unifying formalism for reward learning

**Hong Jun Jeon**[*1]**, Smitha Milli**[*2]**, Anca Dragan**[2]
hjjeon@stanford.edu, smilli@berkeley.edu, anca@berkeley.edu
[*]Equal contribution,
[1]Stanford University,
[2]University of California, Berkeley

## Abstract

It is often difficult to hand-specify what the correct reward function is for a task, so researchers have instead aimed to learn reward functions from human behavior or feedback. The types of behavior interpreted as evidence of the reward function have expanded greatly in recent years. We've gone from demonstrations, to comparisons, to reading into the information leaked when the human is pushing the robot away or turning it off. And surely, there is more to come. How will a robot make sense of all these diverse types of behavior? Our key observation is that different types of behavior can be interpreted in a single unifying formalism - as a *reward-rational choice* that the human is making, often implicitly. We use this formalism to survey prior work through a unifying lens, and discuss its potential use as a recipe for interpreting new sources of information that are yet to be uncovered.

## 1 Introduction

It is difficult to specify reward functions that always lead to the desired behavior. Recent work has argued that the reward specified by a human is merely a source of information about what people actually want a robot to optimize, i.e., the intended reward [1, 2]. Luckily, it is not the only one. Robots can also learn about the intended reward from demonstrations (IRL) [3, 4], by asking us to make comparisons between trajectories [5, 6, 7], or by grounding our instructions [8, 9].

Perhaps even more fortunate is that we seem to *leak* information left and right about the intended reward. For instance, if we push the robot away, this shouldn't just modify the robot's *current* behavior – it should also inform the robot about our preferences more *generally* [10, 11]. If we turn the robot off in a state of panic to prevent it from disaster, this shouldn't just stop the robot right now. It should also inform the robot about the intended reward function so that the robot prevents itself from the same disaster in the future: the robot should infer that whatever it was about to do has a tragically low reward. Even the *current state* of the world ought to inform the robot about our preferences – it is a direct result of us having been acting in the world according to these preferences [12]! For instance, those shoes didn't magically align themselves at the entrance, someone put effort into arranging them that way, so their state alone should tell the robot something about what we want.

Overall, there is much information out there, some purposefully communicated, other leaked. While existing papers are instructing us how to tap into some of it, one can only imagine that there is much more that is yet untapped. There are probably new yet-to-be-invented ways for people to purposefully provide feedback to robots – e.g. guiding them on which part of a trajectory was particularly good or bad. And, there will probably be new realizations about ways in which human behavior already leaks information, beyond the state of the world or turning the robot off. How will robots make sense of all these diverse sources of information?

Our insight is that there is a way to interpret all this information in a single unifying formalism. The critical observation is that human behavior is a *reward-rational implicit choice* – a choice from an implicit set of options, which is approximately rational for the intended reward. This observation leads to a *recipe* for making sense of human behavior, from language to switching the robot off. The recipe has two ingredients: 1) the set of *options* the person (implicitly) chose from, and 2) a *grounding* function that maps these options to robot behaviors. This is admittedly obvious for traditional feedback. In comparison feedback, for instance, the set of options is just the two robot behaviors presented to the human to compare, and the grounding is identity. In other types of behavior though, it is much less obvious. Take switching the robot off. The set of options is implicit: you can turn it off, or you can do nothing. The formalism says that when you turn it off, it should know that you could have done nothing, but (implicitly) chose not to. That, in turn, should propagate to the robot's reward function. For this to happen, the robot needs to ground these options to robot behaviors: identity is no longer enough, because it cannot directly evaluate the reward of an utterance or of getting turned off, but it can evaluate the reward of robot actions or trajectories. Turning the robot off corresponds to a trajectory – whatever the robot did until the off-button was pushed, followed by doing nothing for the rest of the time horizon. Doing nothing corresponds to the trajectory the robot was going to execute. Now, the robot knows you prefer the former to the latter. We have taken a high-level human behavior, and turned it into a direct comparison on robot trajectories with respect to the intended reward, thereby gaining reward information.

We use this perspective to survey prior work on reward learning. We show that despite their diversity, many sources of information about rewards proposed thus far can be characterized as instantiating this formalism (some very directly, others with some modifications). This offers a unifying lens for the area of reward learning, helping better understand and contrast prior methods. We end with discussion on how the formalism can help combine and actively decide among feedback types, and also how it can be a potentially helpful recipe for interpreting new types of feedback or sources of leaked information.

## 2 A formalism for reward learning

### 2.1 Reward-rational implicit choice

In reward learning, the robot's goal is to learn a reward function $r : \Xi \to \mathbb{R}$ from human behavior that maps trajectories[1] $\xi \in \Xi$ to scalar rewards.

**(Implicit/explicit) set of options $\mathcal{C}$.** We interpret human behavior as choosing an option $c^*$ from a set of options $\mathcal{C}$. Different behavior types will correspond to different explicit or implicit sets $\mathcal{C}$. For example, when a person is asked for a *trajectory comparison*, they are explicitly shown two trajectories and they pick one. However, when the person gives a *demonstration*, we think of the possible options $\mathcal{C}$ as implicitly being all possible trajectories the person could have demonstrated. The implicit/explicit distinction brings out a general tradeoff in reward learning. The cleverness of implicit choice sets is that even when we cannot enumerate and show all options to the human, e.g. in demonstrations, we still rely on the human to optimize over the set. On the other hand, an implicit set is also risky – since it is not explicitly observed, we may get it wrong, potentially resulting in worse reward inference.

**The grounding function $\psi$.** We link the human's choice to the reward by thinking of the choice as (approximately) maximizing the reward. However, it is not immediately clear what it means for the human to maximize reward when choosing feedback because the feedback may not be a (robot) trajectory, and the reward is only defined over trajectories. For example, in *language feedback*, the human describes what they want in words. What is the reward of the sentence, "Do not go over the water"?

To overcome this syntax mismatch, we map options in $\mathcal{C}$ to (distributions over) trajectories with a grounding function $\psi : \mathcal{C} \to f_\Xi$ where $f_\Xi$ is the set of distributions over trajectories for the robot $\Xi$. Different types of feedback will correspond to different groundings. In some instances, such as kinesthetic demonstrations or trajectory comparisons, the mapping is simply the identity. In others, like corrections, language, or proxy rewards, the grounding is more complex (see Section 3).

**Human policy.** Given the set of choices $\mathcal{C}$ and the grounding function $\psi$, the human's approximately rational choice $c^* \in \mathcal{C}$ can now be modeled via a *Boltzmann-rational* policy, a policy in which the probability of choosing an option is exponentially higher based on its reward:

$$\mathbb{P}(c^* \mid r, \mathcal{C}) = \frac{\exp(\beta \cdot \mathbb{E}_{\xi \sim \psi(c^*)}[r(\xi)])}{\sum_{c \in \mathcal{C}} \exp(\beta \cdot \mathbb{E}_{\xi \sim \psi(c)}[r(\xi)])}, \tag{1}$$

where the parameter $\beta$ is a coefficient that models how rational the human is. Often, we simplify Equation 1 to the case where $\psi$ is a deterministic mapping from choices in $\mathcal{C}$ to trajectories in $\Xi$, instead of distributions over trajectories. Then, the probability of choosing $c^*$ can be written as:[2]

$$\mathbb{P}(c^* \mid r, \mathcal{C}) \propto \exp(\beta \cdot r(\psi(c^*))) \tag{2}$$

Boltzmann-rational policies are widespread in psychology [13, 14, 15], economics [16, 17, 18, 17], and AI [19, 20, 21, 22, 23] as models of human choices, actions, or inferences. But why are they a reasonable model?

While there are many possible motivations, we contribute a derivation (Appendix A) as the maximum-entropy distribution over choices for a *satisficing* agent, i.e. an agent that in expectation makes a choice with $\epsilon$-optimal reward. A higher value of $\epsilon$ results in a lower value of $\beta$, modeling less optimal humans.

**Definition 2.1** (Reward-rational choice). Finally, putting it all together, we call a type of feedback a *reward-rational choice* if, given a grounding function $\psi$, it can be modeled as a choice from an (explicit or implicit set) $\mathcal{C}$ that (approximately) maximizes reward, i.e., as in Equation 1.

## 2.2 Robot inference

Each feedback is an observation about the reward, which means the robot can run Bayesian inference to update its belief over the rewards. For a determinstic grounding,

$$\mathbb{P}(r \mid c^*) = \frac{1}{Z} \cdot \frac{\exp(\beta \cdot r(\psi(c^*)))}{\sum_{c \in \mathcal{C}} \exp(\beta \cdot r(\psi(c)))} \cdot \mathbb{P}(r), \tag{3}$$

where $\mathbb{P}(r)$ is the prior over rewards and $Z$ is the normalization over possible reward functions. The inference above is often intractable, and so reward learning work leverages approximations [24], or computes only the MLE for a parametrization of rewards (more recently as weights in a neural network on raw input [7, 25]).

Finally, when the human is highly rational ($\beta \to \infty$), the only choices in $\mathcal{C}$ with a non-negligible probability of being picked are the choices that exactly maximize reward. Thus, the human's choice $c^*$ can be interpreted as *constraints* on the reward function (e.g. [26]):

$$\text{Find } r \text{ such that } r(\psi(c^*)) \geq r(\psi(c)) \quad \forall c \in \mathcal{C}. \tag{4}$$

## 3 Prior work from the perspective of the formalism

We now instantiate the formalism above with different behavior types from prior work, constructing their choice sets $\mathcal{C}$ and groundings $\psi$. Some are obvious – comparisons, demonstrations especially. Others – initial state, off, reward/punish – are more subtle and it takes slightly modifying their original methods to achieve unification, speaking to the nontrivial nuances of identifying a common formalism.

Table 1 lists $\mathcal{C}$ and $\psi$ for each feedback, while Table 2 shows the deterministic constraint on rewards each behavior imposes, along with the probabilistic observation model – highlighting, despite the differences in feedback, the pattern of the (exponentiated) choice reward in the numerator, and the normalization over $\mathcal{C}$ in the denominator. Fig. 1 will serve as the illustration for these types,

Table 1: The choice set $\mathcal{C}$ and grounding function $\psi$ for different types of feedback described in Section 3, unless otherwise noted.

| Feedback | Choices $\mathcal{C}$ | Grounding $\psi$ |
|---|---|---|
| Comparisons [5] | $\xi_i \in \{\xi_1, \xi_2\}$ | $\psi(\xi_i) = \xi_i$ |
| Demonstrations [3] | $\xi_d \in \Xi$ | $\psi(\xi) = \xi$ |
| Corrections [11] | $\Delta q \in Q - Q$ | $\psi(\Delta q) = \xi_R + A^{-1}\Delta q$ |
| Improvement [10] | $\xi \in \{\xi_{\text{improved}}, \xi_R\}$ | $\psi(\xi) = \xi$ |
| Off [27] | $c \in \{\text{off}, -\}$ | $\psi(c) = \begin{cases} \xi_R & c = - \\ \xi_R^{0:t}\xi_R^t \ldots \xi_R^t & c = \text{off} \end{cases}$ |
| Language [28] | $\lambda \in \Lambda$ | $\psi(\lambda) = \text{Unif}(G(\lambda))$ |
| Proxy Rewards [1] | $\tilde{r} \in \tilde{\mathcal{R}}$ | $\psi(\tilde{r}) = \pi(\xi \mid \tilde{r})$ |
| Reward and Punishment [29] | $c \in \{+1, -1\}$ | $\psi(c) = \begin{cases} \xi_R & c = +1 \\ \xi_{\text{expected}} & c = -1 \end{cases}$ |
| Initial state [12] | $s \in \mathcal{S}$ | $\psi(s) = \text{Unif}(\{\xi_H^{-T:0} \mid \xi_H^0 = s\})$ |
| Credit assignment (Discussion) | $\xi \in \{\xi_R^{i:i+k}, 0 \le i \le T\}$ | $\psi(\xi) = \xi$ |

Table 2: The probabilistic model (Equation 1) and the simplification to the constraint-based model (Equation 4).

| Feedback | Constraint | Probabilistic |
|---|---|---|
| Comparisons | $r(\xi_1) \ge r(\xi_2)$ | $\mathbb{P}(\xi_1 \mid r, \mathcal{C}) = \dfrac{\exp(\beta \cdot r(\xi_1))}{\exp(\beta \cdot r(\xi_1)) + \exp(\beta \cdot r(\xi_2))}$ |
| Demonstrations | $r(\xi_D) \ge r(\xi) \quad \forall \xi \in \Xi$ | $\mathbb{P}(\xi_D \mid r, \Xi) = \dfrac{\exp(\beta \cdot r(\xi_D))}{\sum_{\xi \in \Xi} \exp(\beta \cdot r(\xi))}$ |
| Corrections | $r(\xi_R + A^{-1}\Delta q) \ge r(\xi_R + A^{-1}\Delta q') \, \forall \Delta q' \in Q - Q$ | $\mathbb{P}(\Delta q' \mid r, Q - Q) = \dfrac{\exp(\beta \cdot r(\xi_R + A^{-1}\Delta q))}{\sum_{\Delta q \in Q-Q} \exp(\beta \cdot r(\xi_R + A^{-1}\Delta q))}$ |
| Improvement | $r(\xi_{\text{improved}}) \ge r(\xi_R)$ | $\mathbb{P}(\xi_{\text{improved}} \mid r, \mathcal{C}) = \dfrac{\exp(\beta \cdot r(\xi_{\text{improved}}))}{\exp(\beta \cdot r(\xi_{\text{improved}})) + \exp(\beta \cdot r(\xi_R))}$ |
| Off | $r(\xi_R^{0:t}\xi^t \ldots \xi^t) \ge r(\xi_R)$ | $\mathbb{P}(\text{off} \mid r, \mathcal{C}) = \dfrac{\exp(\beta \cdot r(\xi_R^{0:t}\xi^t \ldots \xi^t))}{\exp(\beta \cdot r(\xi_R^{0:t}\xi^t \ldots \xi^t)) + \exp(\beta \cdot r(\xi_R))}$ |
| Language | $\mathbb{E}_{\xi \sim \text{Unif}(G(\lambda^*))}[r(\xi)] \ge \mathbb{E}_{\xi \sim \text{Unif}(G(\lambda))}[r(\xi)] \, \forall \lambda \in \Lambda$ | $\mathbb{P}(\lambda^* \mid r, \Lambda) = \dfrac{\exp(\beta \cdot \mathbb{E}_{\xi \sim \text{Unif}(G(\lambda^*))}[r(\xi)])}{\sum_{\lambda \in \Lambda} \exp(\beta \cdot \mathbb{E}_{\xi \sim \text{Unif}(G(\lambda))}[r(\xi)])}$ |
| Proxy Rewards | $\mathbb{E}_{\tilde{\xi} \sim \pi(\tilde{\xi}\mid\tilde{r})}[r(\tilde{\xi})] \ge \mathbb{E}_{\tilde{\xi} \sim \pi(\tilde{\xi}\mid c)}[r(\tilde{\xi})] \quad \forall c \in \tilde{\mathcal{R}}$ | $\mathbb{P}(\tilde{r} \mid r, \tilde{\mathcal{R}}) = \dfrac{\exp(\beta \cdot \mathbb{E}_{\tilde{\xi} \sim \pi(\tilde{\xi}\mid\tilde{r})}[r(\tilde{\xi})])}{\sum_{c \in \tilde{\mathcal{R}}} \exp(\beta \cdot \mathbb{E}_{\tilde{\xi} \sim \pi(\tilde{\xi}\mid c)}[r(\tilde{\xi})])}$ |
| Reward/Punish | $r(\xi_R) \ge r(\xi_{\text{expected}})$ | $\mathbb{P}(+1 \mid r, \mathcal{C}) = \dfrac{\exp(\beta \cdot r(\xi_R))}{\exp(\beta \cdot r(\xi_R)) + \exp(\beta \cdot r(\xi_{\text{expected}}))}$ |
| Initial state | $\mathbb{E}_{\xi \sim \psi(s^*)}[r(s^*)] \ge \mathbb{E}_{\xi \sim \psi(s)}[r(s)] \quad \forall s \in \mathcal{S}$ | $\mathbb{P}(s^* \mid r, \mathcal{S}) = \dfrac{\exp(\beta \cdot \mathbb{E}_{\xi \sim \psi(s^*)}[r(\xi)])}{\sum_{s \in S} \exp(\beta \cdot \mathbb{E}_{\xi \sim \psi(s)}[r(\xi)])}$ |
| Meta-choice | $\mathbb{E}_{\xi \sim \psi(\mathcal{C}_i)}[r(\xi)] \ge \mathbb{E}_{\xi \sim \psi(\mathcal{C}_j)}[r(\xi)] \quad \forall j \in [n]$ | $\mathbb{P}(\mathcal{C}_i \mid r, \mathcal{C}_0) = \dfrac{\exp(\beta_0 \cdot \mathbb{E}_{\xi \sim \psi_0(\mathcal{C}_i)}[r(\xi)])}{\sum_{j \in [n]} \exp(\beta_0 \cdot \mathbb{E}_{\xi \sim \psi_0(\mathcal{C}_j)}[r(\xi)])}$ |
| Credit assignment | $r(\xi^*) \ge r(\xi) \quad \forall \xi \in \mathcal{C}$ | $\mathbb{P}(\xi^* \mid r, \mathcal{C}) = \dfrac{\exp(\beta \cdot r(\xi^*))}{\sum_{\xi \in \mathcal{C}} \exp(\beta \cdot r(\xi))}$ |

looking at a grid world navigation task around a rug. The space of rewards we use for illustration is three-dimensional weight vectors for avoiding the rug, not getting dirty, and reaching the goal.

**Trajectory comparisons.** In trajectory comparisons [30], the human is typically shown two trajectories $\xi_1 \in \Xi$ and $\xi_2 \in \Xi$, and then asked to select the one that they prefer. They are perhaps the most obvious exemplar of reward-rational choice: the set of choices $\mathcal{C} = \{\xi_1, \xi_2\}$ is explicit, and the grounding $\psi$ is simply the identity. As Fig. 1 shows, for linear reward functions, a comparison corresponds to a hyperplane that cuts the space of feasible reward functions in half. For all the reward functions left, the chosen trajectory has higher reward than the alternative. Most work on comparisons is done in the preference-based RL domain in which the robot might compute a policy directly to agree with the comparisons, rather than explicitly recover the reward function [31, 32]. Within methods that do recover rewards, most use the constraint version (left column of Table 2)

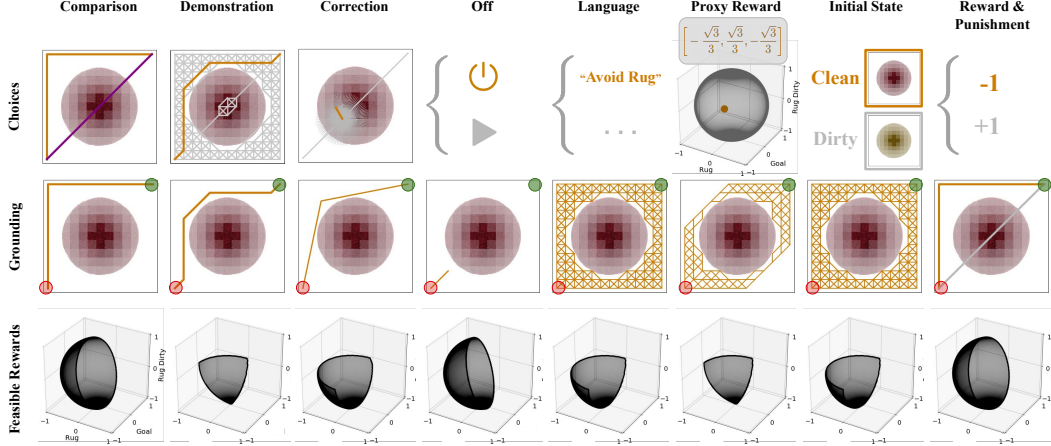

Figure 1: Different behavior types described in Sec. 3 in a gridworld with three features: avoiding/going on the rug, getting the rug dirty, and reaching the goal (green). For each, we display the choices, grounding, and feasible rewards under the constraint formulation of robot inference (4). Each trajectory is a finite horizon path that begins at the start (red). Orange is used to denote $c^*$ and $\psi(c^*)$ while gray to denote other choices $c$ in $\mathcal{C}$. For instance, the **comparison** affects the feasible reward space by removing the halfspace where going on the rug is good. It does not inform the robot about the goal, because both end at the goal. The **demonstration** removes the space where the rug is good, where the goal is bad (because alternates do not reach the goal), and where getting the rug dirty is good (because alternates slightly graze the rug). The **correction** is similar to the demonstration, but does not infer about the goal, since all corrections end at goal.

using various losses [33, 34]. [35] uses the Boltzmann model (right column of Table 2) and proposes actively generating the queries, [6] follows up with actively synthesizing the queries from scratch, and [7] introduces deep neural network reward functions.

**Demonstrations.** In demonstrations, the human is asked to demonstrate the optimal behavior. Reward learning from demonstrations is often called *inverse reinforcement learning* (IRL) and is one of the most established types of feedback for reward learning [3, 4, 19]. Unlike in comparisons, in demonstrations, the human is not explicitly given a set of choices. However, we assume that the human is *implicitly* optimizing over all possible trajectories (Fig. 1 (1st row, 2nd column) shows these choices in gray). Thus, demonstrations are a reward-rational choice in which the set of choices $\mathcal{C}$ is (implicitly) the set of trajectories $\Xi$. Again, the grounding $\psi$ is the identity. In Fig. 1, fewer rewards are consistent with a demonstration than with a comparison. Early work used the constraint formulation with various losses to penalize violations [3, 26]. Bayesian IRL [20] exactly instantiates the formalism using the Boltzmann distribution by doing a full belief update as in Equation 3. Later work computes the MLE instead [19, 22, 36] and approximates the partition function (the denominator) by a quadratic approximation about the demonstration [37], a Laplace approximation [38], or importance sampling [21].

**Corrections** are the first type of feedback we consider that has both an implicit set of choices $\mathcal{C}$ and a non-trivial (not equal to identity) grounding. Corrections are most common in physical human-robot interaction (pHRI), in which a human physically corrects the motion of a robot. The robot executes a trajectory $\xi_R$, and the human intervenes by applying a correction $\Delta q \in Q$ that modifies the robot's current configuration. Therefore, the set of choices $\mathcal{C} = Q - Q$ consists of all possible configuration differences $\Delta q$ the person could have used (Fig. 1 1st row, 3rd column shows possible $\Delta q$s in gray and the selected on in orange). The way we can ground these choices is by finding a trajectory that is closest to the original, but satisfies the constraint of matching a new point:

$$\min_{\xi} ||\xi - \xi_R||^2_A$$
$$s.t.\ \xi(0) = \xi_R(0),\ \xi(T) = \xi_R(T), \xi(t) = \xi_R(t) + \Delta q \tag{5}$$

where $t$ is the time at which the correction was applied. Choosing a non-Euclidean inner-product, A (for instance $K^T K$, with $K$ the finite differencing matrix), couples states along the trajectory in time and leads to a the resulting trajectory smoothly deforming – propagating the change $\Delta q$ to

the rest of the trajectory: $\psi(\Delta q) = \xi_R + A^{-1}[\lambda, 0, .., \Delta q, .., 0, \gamma]^T$ (with $\lambda$ and $\gamma$ making sure the end-points stay in place). This is the orange trajectory in the figure. Most work in corrections affects the robot's trajectory but not the reward function [39, 40], with [41] proposing the propagation via $A^{-1}$ above. [11] propose that corrections are informative about the reward and use the propagation as their grounding, deriving an approximate MAP estimate for the reward. [42] introduce a way to maintain uncertainty.

**Improvement.** Prior work [10] has also modeled a variant of corrections in which the human provides an improved trajectory $\xi_{improved}$ which is treated as better than the robot's original $\xi_R$. Although [10] use the Euclidean inner product and implement reward learning as an online gradient method that treats the improved trajectory as a demonstration (but only takes a single gradient step towards the MLE), we can also naturally interpret improvement as a comparison that tells us the improved trajectory is better than the original: the set of options $\mathcal{C}$ consists of only $\xi_R$ and $\xi_{improved}$ now, as opposed to all the trajectories obtainable by propagating local corrections; the grounding is identity, resulting in essentially a comparison between the robot's trajectory and the user provided one.

**Off.** In "off" feedback, the robot executes a trajectory, and at any point, the human may switch the robot off. "Off" appears to be a very sparse signal, and it is not spelled out in prior work how one might learn a reward from it. Reward-rational choice suggests that we first uncover the implicit set of options $\mathcal{C}$ the human was choosing from. In this case, the set of options consists of turning the robot off or not doing anything at all: $\mathcal{C} = \{\text{off}, -\}$. Next, we must ask how to evaluate the reward of the two options, i.e., what is the grounding? [27] introduced off feedback and formalized it as a choice for a one-shot game. There, not intervening means the robot takes its one possible action, and intervening means the robot takes the no-op action. This can be easily generalized to the sequential setting: not intervening means that the robot continues on its current trajectory, and intervening means that it stays at its current position for the remainder of the time horizon. Thus, the choices $\mathcal{C} = \{\text{off}, -\}$ map to the trajectories $\{\xi_R^{0:t}\xi_R^t \ldots \xi_R^t, \xi_R\}$.

**Language.** Humans might use rich language to instruct the robot, like "Avoid the rug." Let $G(\lambda)$ be the trajectories that are consistent with an utterance $\lambda \in \Lambda$ (e.g. all trajectories that do not enter the rug). Usually the human instruction is interpreted *literally*, i.e. any trajectory consistent with the instruction $\xi \in G(\lambda)$ is taken to be equally likely , although, other distributions are also possible. For example, a problem with literal interpretation is that it does not take into account the other choices the human may have considered. The instruction "Do not go into the water" is consistent with the robot not moving at all, but we imagine that if the human wanted the robot to do nothing, they would have said that instead. Therefore, it would be incorrect for the robot to do nothing when given the instruction "Do not go into the water". This type of reasoning is called *pragmatic reasoning*, and indeed recent work shows that explicitly interpreting instructions pragmatically can lead to higher performance [43, 44]. The reward-rational choice formulation of language feedback naturally leads to pragmatic reasoning on the part of the robot, and is in fact equivalent to the rational speech acts model [15], a standard model of pragmatic reasoning in language. The pragmatic reasoning arises because the human is explicitly modeled as choosing from a set of options.

The reward-rational choice formulation of language feedback naturally leads to pragmatic reasoning on the part of the robot, and is in fact equivalent to the rational speech acts model [15], a standard model of pragmatic reasoning in language. The pragmatic reasoning arises because the human is explicitly modeled as choosing from a set of options. Language is a reward-rational choice in which the set of options $\mathcal{C}$ is the set of instructions considered in-domain $\Lambda$ and the grounding $\psi$ maps an utterance $\lambda$ to the uniform distribution over consistent trajectories $\text{Unif}(G(\lambda))$. In language feedback, a key difficulty is learning which robot trajectories are consistent with a natural language instruction, the *language grounding problem* (and is where we borrow the term "grounding" from) [28, 45, 9]. Fig. 1 shows the grounding for avoiding the rug in orange – all trajectories from start to goal that do not enter rug cells.

**Proxy rewards** are expert-specified rewards that do not necessarily lead to the desired behavior in all situations, but can be trusted on the training environments. They were introduced by [1], who argued that even when the expert attempts to fully specify the reward, it will still fail to generalize to some situations outside of the training environments. Therefore, rather than taking a specified reward at face value, we can interpret it as evidence about the true reward. Proxy reward feedback is a reward-rational choice in which the set of choices $\mathcal{C}$ is the set of proxy rewards the designer may have chosen, $\tilde{\mathcal{R}}$. The reward designer is assumed to be approximately-optimal, i.e. they are more

likely to pick a proxy reward $\tilde{r} \in \tilde{\mathcal{R}}$ if it leads to better trajectories *on the training environment(s)*. Thus, the grounding $\psi$ maps a proxy reward $r$ to the distribution over trajectories that the robot takes in the training environment given the proxy reward [1, 46, 2]. Fig. 1 shows the grounding for a proxy reward for reaching the goal, avoiding the rug, and not getting the rug dirty – many feasible rewards would produce similar behavior as the proxy. By taking the proxy as evidence about the underlying reward, the robot ends up with uncertainty over what the actual reward might be, and can better hedge its bets at test time.

**Reward and punishment** [29, 47]. In this type of feedback, the human can either reward (+1) or punish (−1) the robot for its trajectory $\xi_R$; the set of options is $\mathcal{C} = \{+1, -1\}$. A naive implementation would interpret reward and punishment literally, i.e. as a scalar reward signal for a reinforcement learning agent, however empirical studies show that humans reward and punish based on how well the robot performs *relative to their expectations* [48]. Thus, we can use our formalism to interpret that: reward (+1) grounds to the robot's trajectory $\xi_R$, while punish (−1) grounds to the trajectory the human expected $\xi_{\text{expected}}$ (not necessarily observed).

**Initial state.** Shah et al. [12] make the observation that when the robot is deployed in an environment that humans have acted in, the current state of the environment is already optimized for what humans want, and thus contains information about the reward. For example, suppose the environment has a goal state which the robot can reach through either a paved path or a carpet. If the carpet is pristine and untrodden, then humans must have intentionally avoided walking on it in the past (even though the robot hasn't observed this past behavior), and the robot can reasonably infer that it too should not go on the carpet.

The original paper inferred rewards from a single state $s$ by marginalizing over possible pasts, i.e. trajectories $\xi_H^{-T:0}$ that end at $s$ which the human could have taken, $P(s|r) = \sum_{\xi_H^{-T:0}|\xi_H(0)=s} P(\xi_H^{-T:0}|r)$. However, through the lens of our formalism, we see that initial states can also be interpreted more directly as reward-rational implicit choices. The set of choices $\mathcal{C}$ can be the set of possible initial states $\mathcal{S}$. The grounding function $\psi$ maps a state $s \in \mathcal{S}$ to the uniform distribution over any human trajectory $\xi_H^{-T:0}$ that starts from a specified time before the robot was deployed ($t = -T$) and ends at state $s$ at the time the robot was deployed ($t = 0$), i.e. $\xi_H^0 = s$. This leads to the $P(s|r)$ from Table 2, which is almost the same as the original, but sums over trajectories directly in the exponent, and normalizes over possible other states. The two interpretations would only become equivalent if we replaced the Boltzmann distribution with a linear one. Fig 1 shows the result of this (modified) inference, recovering as much information as with the correction or language.

## 4    Discussion of implications

From demonstrations to reward/punishment to the initial state of the world, the robot can extract information from humans by modeling them as making approximate reward-rational choices. Often, the choices are implicit, like in turning the robot off or providing language instructions. Sometimes, the choices are not made in order to purposefully communicate about the reward, and rather end up leaking information about it, like in the initial state, or even in corrections or turning the robot off. Regardless, this unifying lens enables us to better understand, as in Fig. 1, how all these sources of information relate and compare.

Down the line, we hope this formalism will enable research on combining and actively querying for feedback types, as well as making it easier to do reward learning from new, yet to be uncovered sources of information. Concretely, so far we have talked about learning from individual types of behaviors. But we do not want our robots stuck with a single type: we want them to 1) read into all the leaked information, and 2) learn from all the purposeful feedback. For example, the robot might receive demonstrations from a human during training, and then corrections during deployment, which were followed by the human prematurely switching the robot off. The observational model in (2) for a single type of behavior also provides a natural way to model combinations of behavior. If each observation is conditionally independent given the reward, then according to (2), the probability of observing a vector $\mathbf{c}$ of $n$ behavioral signals (of possibly different types) is equal to

$$\mathbb{P}(\mathbf{c} \mid r) = \prod_{i=1}^{n} \frac{\exp(\beta_i \cdot r(\psi_i(\mathbf{c}_i)))}{\sum_{c \in \mathcal{C}_i} \exp(\beta_i \cdot r(\psi_i(c)))} . \tag{6}$$

Given this likelihood function for the human's behavior, the robot can infer the reward function using the approaches and approximations described in Sec. 2.2. Recent work has already built in this direction, combining trajectory comparisons and demonstrations [25, 49]. We note that the formulation in Equation 6 is general and applies to *any* combination. In Appendix B, we describe a case study on a novel combination of feedback types: proxy rewards, a physical improvement, and comparisons in which we use a constraint-based approximation (see Equation 4) to Equation 6.

Further, it also becomes natural to *actively decide* which feedback type to ask a human for. Rather than relying on a heuristic (or on the human to decide), the robot can maximize expected information gain. Suppose we can select between $n$ types of feedback with choice sets $\mathcal{C}_1, \ldots, \mathcal{C}_n$ to ask the user for. Let $b_t$ be the robot's belief distribution over rewards at time $t$. The type of feedback $i^*$ that (greedily) maximizes information gain for the next time step is

$$i^* = \operatorname*{arg\,max}_{i \in [n]} \mathbb{E}_{r_t, c_i^*} \left[ \log \left( \frac{p(c_i^* \mid r_t)}{\int_{r_t \in \mathcal{R}} p(c_i^* \mid r_t) b_t(r_t)} \right) \right] , \qquad (7)$$

where $r_t \sim B_t$ is distributed according to the robot's current belief, $c_i^* \in \mathcal{C}_i$ is the random variable corresponding to the user's choice within feedback type $i$, and $p(c_i^* \mid r_t)$ is defined according to the human model in Equation 1. We also note that different feedback types may have different costs associated with them (e.g. of human time) and it is straight-forward to integrate these costs into (7). In Appendix C, we describe experiments with active selection of feedback types. In the environments we tested, we found that demonstrations are optimal early on, when little is known about the reward, while comparisons became optimal later, as a way to fine-tune the reward. The finding provides validation for the approach pursued by [49] and [25]. Both papers manually define the mixing procedure we found to be optimal: initially train the reward model using human demonstrations, and then fine-tune with comparisons.

Finally, the types of feedback or behavior we have discussed so far are by no means the only types possible. New ones will inevitably be invented. But when designing a new type of feedback, it is often difficult to understand what the relationship is between the reward $r$ and the feedback $c^*$. Reward-rational choice suggests a recipe for uncovering this link – define what the implicit set of options the human is choosing from is, and how those options ground to trajectories. Then, Equation 1 provides a formal model for the human feedback.

For example, hypothetically, someone might propose a "credit assignment" type of feedback. Given a trajectory $\xi_R$ of length $T$, the human is asked to pick a segment of length $k < T$ that has maximal reward. We doubt the set of choices in an implementation of credit assignment would be explicit, however the implicit set of choices $\mathcal{C}$ is then the set of all segments of length $k$. The grounding function $\psi$ is simply the identity. With this choice of $\mathcal{C}$ and $\psi$ in hand, the human can now be modeled according to Equation 1, as we show in the last rows of Tables 1 and 2.

While of course the formalism won't apply to *all* types of feedback, we believe that it applies to *many*, even to types that initially seem to have a more obvious, literal interpretation (e.g. reward and punishment, Section 3). Most immediately, we are excited about using it to formalize a particular new source of (leaked) information we uncovered while developing the formalism itself: the moment we enable robots to learn from multiple types of feedback, users will have the *choice* of which feedback to provide. Interpreted literally, each feedback gives the robot evidence about the reward. However, this leaves information on the table: if the person decided to, say, turn the robot off, they *implicitly* decided to *not* provide a correction, or use language. Intuitively, this means that turning off the robot was a more appropriate intervention with respect to the true reward. Interpreting the feedback type itself as reward-rational implicit choice has the potential to enable robots to extract more information about the reward from the same data. We call the choice of feedback type "meta-choice". In Appendix D, we formalize meta-choice and conduct experiments that showcase its potential importance.

Overall, we see this formalism as providing conceptual clarity for existing and future methods for learning from human behavior, and a fruitful base for future work on multi-behavior-type reward learning.

## Broader Impact

As AI capability advances, it is becoming increasingly important to align the objectives of AI agents to what people want. From how assistive robots can best help their users, to how autonomous cars should trade off between safety risk and efficiency, to how recommender systems should balance revenue considerations with longer-term user happiness and with avoiding influencing user views, agents cannot rely on a reward function specified once and set in stone. By putting different sources of information about the reward explicitly under the same framework, we hope our paper contributes towards a future in which agents maintain uncertainty over what their reward should be, and use different types of feedback from humans to refine their estimate and become better aligned with what people want over time – be them designers or end-users.

On the flip side, changing reward functions also raises its own set of risks and challenges. First, the relationship between designer objectives and end-user objectives is not clear. Our framework can be used to adapt agents to end-users preferences, but this takes away control from the system designers. This might be desirable for, say, home robots, but not for safety-critical systems like autonomous cars, where designers might need to enforce certain constraints a-priori on the reward adaptation process. More broadly, most systems have multiple stake-holders, and what it means to do ethical preference aggregation remains an open problem. Further, if the robot's model of the human is misspecified, adaptation might lead to more harm than good, with the robot inferring a worse reward function than what a designer could specify by hand.

## Acknowledgments and Disclosure of Funding

We thank the members of the InterACT lab for fruitful discussion and advice, especially Dylan Hadfield-Menell for his perspectives on the relationship between demonstrations and comparisons. We thank Andreea Bobu, Paul Christiano, and Rohin Shah for their feedback on the manuscript.

This work is partially supported by ONR YIP and Open Philanthropy Project. This material is based upon work supported by the National Science Foundation Graduate Research Fellowship under Grant No. 1752814. Any opinion, findings, and conclusions or recommendations expressed in this material are those of the authors(s) and do not necessarily reflect the views of the National Science Foundation.

## Footnotes

[1]We consider finite fixed horizon T trajectories.

[2]One can also consider a variant of Equation 1 in which choices are grounded to actions, rather than trajectories, and are evaluated via a Q-value function, rather than the reward function. That is, $\psi : \mathcal{C} \to \mathcal{A}$, $\mathbb{P}(c^* \mid r, \mathcal{C}) \propto \exp(\beta \cdot \mathbb{E}_{a \sim \psi(c^*)}[Q^*(s, a)])$.

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
