[Supplementary Material]

# A  Bounded rationality, maximum entropy, and Boltzmann-rational policies

A perspective on reward learning that makes use at its core the Boltzmann model from Equation 1 would not be complete without a formal justification for it within our context. In this section, we derive it as the maximum-entropy distribution for the choices made by a bounded, *satisficing* human. Our explanation is complementary to that of [50] who derive an axiomatic, thermodynamic framework to modeling bounded-rational decision making. Their framework leads to much the same interpretation of the Boltzmann-rational distribution, but is significantly more complex than needed for our purposes.

A perfectly rational human choosing from the set $\mathcal{C}$ would always pick the choice with optimal reward, $\max_{c\in\mathcal{C}} r(\psi(c))$. However, since humans are bounded, we do not expect them to perform optimally. Herbert Simon proposed the influential idea that humans are *bounded rational* and merely *satisfice* [51], rather than maximize, i.e., they pick an option above some satisfactory threshold, rather than picking the best possible option.

We can abstractly model a satisficing human by modeling their expected reward as equal to a satisficing threshold, $\max_{c\in\mathcal{C}} r(\psi(c)) - \epsilon$ where $\epsilon \in (0, \epsilon_{\max})$ is the amount of expected error. The maximum possible error, $\epsilon_{max} = \min_{c\in\mathcal{C}} r(\psi(c)) - \max_{c\in\mathcal{C}} r(\psi(c))$, corresponds to *anti-rationality*, i.e., always picking the worst option.

Given the constraint that the human's expected reward is satisfactory, how should we pick a distribution to model the human's choices? The principle of maximum entropy [52] gives us a guide. If we want to encode no extra information in the distribution, then we ought to pick the distribution that maximizes entropy subject to the constraint on the satisficing threshold.

**Definition A.1** (Satisficing MaxEnt problem). Let be be a distribution $P$ over choice set $\mathcal{C}$ and let $p$ be a density for $P$ with respect to a base measure $F$. The Shannon entropy of $P$ is defined as $H(P) = - \int_{\mathcal{C}} p(f) \log p(f) dF(f)$. The *satisficing maximum entropy problem* is to find a distribution $P$ that maximizes entropy subject to the satisficing constraint (8):

$$\max_P H(P)$$

$$\text{subject to}$$

$$\mathbb{E}_{c\sim P}[r(\psi(c))] = \max_{c\in\mathcal{C}} r(\psi(c)) - \epsilon\,. \tag{8}$$

It is well-known that the maximum-entropy distribution subject to linear constraints (such as a constraint on the mean like in (8)) is the unique exponential distribution that satisfies the constraints. Thus, for our special case, the maximum-entropy distrbution is the Boltzmann distribution with rationality coefficient $\beta$ satisfying the satisficing constraint.

**Theorem A.1** ([52]). *The solution to the satisficing maximum entropy problem is the Boltzmann distribution* $\mathbb{P}_\beta(f) \propto \exp(\beta \cdot r(\psi(c)))$ *where $\beta$ is the unique value satisfying the satisficing constraint (8).*

Since the expected reward $\mathbb{E}_\beta[r(\psi(c)]$ is monotonically increasing in the rationality parameter $\beta$, the satisficing error $\epsilon$ and rationality coefficient $\beta$ have a one-to-one relationship, as summarized in the following corollary.

**Corollary A.1.** *The solution to the satisficing maximum entropy problem is a Boltzmann-rational policy where the rationality coefficient $\beta$ is monotonically decreasing in the satisficing error $\epsilon$. In particular, we have the following:*

| *Human type* | *Error* $\epsilon$ | *Rationality* $\beta$ |
|---|---|---|
| *Perfectly rational* | $\epsilon \to 0$ | $\beta \to +\infty$ |
| *Random* | $\epsilon = \max_{c\in\mathcal{C}} r(\psi(c)) - \mathbb{E}_{c\sim Unif(\mathcal{C})}[r(\psi(c))]$ | $\beta = 0$ |
| *Anti-rational* | $\epsilon \to \epsilon_{max}$ | $\beta \to -\infty$ |

Thus, we see that the idea of bounded rationality, as in satisficing, and Boltzmann-rationality are in fact equivalent. By following the principle of maximum entropy, Boltzmann-rationality provides a way to model a satisficing human, without implicitly adding in any other assumptions about the human's choice.

Figure 2: A case study for teaching a reward for robot arm motion using two training environments. The robot trades off efficiency, keeping distance away from the human, and also from the table. We use the constraints interpretation of feedback in this study. We start by defining a proxy reward that produces acceptable behavior (orange trajectories) in the training environments (1st row). This initial feedback significantly prunes the feasible space, but is not enough to guarantee good performance in other environments. On the right, we see trajectories still considered feasible in two test environments. The green one is correct, however, the other feasible trajectories are either too close to the human or too close to the robot. After an improvement feedback and a comparison, the robot shrinks the space of feasible rewards, removing extraneous rewards that produce undesirable behavior at test time.

## B   A case study on combining feedback types

Fig. 2 illustrates a case study for teaching a robot arm a reward for motion planning through a novel combination of feedback types. In each environment, the robot arm must plan a trajectory from a start configuration to a designated goal configuration. We want this trajectory to properly trade off efficiency against staying at an appropriate distances to the human, and to the table. Hand-tuning a reward function that returns desirable trajectories in *all* possible environments is actually very challenging. You could imagine that as you increase the efficiency weight to produce a smoother trajectory in one environment, you break the behavior in another environment where the robot now gets too close to the human, etc. In fact, the first type of feedback in the case study illustrates this: we design a (proxy) reward function that works well in two (training) environments (top left), but there are many rewards that are consistent with that behavior, yet produce vastly different behaviors in the two test environments (right).

Therefore, we start by defining a proxy reward, but then follow it up with more feedback: an improvement, and a comparison between two trajectories. This narrows down the space of rewards such that the robot can now generalize what to do outside of the training environments, as shown by two testing environments (right).

**Cost Function and Features.**

$$\text{Efficiency} := \sum_{i=1}^{|\tau|-1} \|\tau[i] - \tau[i-1]\|_2^2 \,,$$

$$\text{Distance to Table} := \sum_{i=0}^{|\tau|-1} 1 - \exp(-\text{dist}_i)$$

$$\text{where dist}_i = |g_{st}(\tau[i])_z - \text{table}_z|,$$

$$\text{Distance to Human} := \sum_{i=0}^{|\tau|-1} 1 - \exp(-\text{dist}_i)$$

$$\text{where dist}_i = \|\text{proj}_x(g_{st}(\tau[i])) - \text{proj}_x(\text{human})\|_2^2.$$

Efficiency is the sum of squared configuration space distances between consecutive trajectory waypoints. The table and human features are expressed as 1 minus a radial basis function of a modified distance between the object and the robot's end effector positon (denoted by $g_{st}(\tau[i])$, where $g_{st}$ is the forward kinematics that maps configuration $\xi[i] \in Q$ to its end effector location in $\mathbb{R}^3$). For the table, this modification is to only consider distance in the z-coordinate, effectively measuring the distance from the robot's end effector to the table *plane*. For the human, the modification is to treat the human as an axis $x$ and consider distance in 2 dimensions after projecting onto the plane with normal $x$. In Figure 2, the main obstacle is either the human's body or his arm. When the body is the obstacle, $x = [0, 0, 1]$ and when the arm is the obstacle, $x = [0, 1, 0]$. This considers the human not as just a point, but rather a line along the body, or arm axis.

**Optimization** We approximate the space of reward parameters $\Theta$ by uniform discretization at the surface of the non-negative octant of the 3 dimensional sphere (1371 points). Robotic motion planners cannot, in general, compute the globally optimal trajectory for a given $\theta \in \Theta$ so we resort to computing a set $\hat{\mathcal{T}}$ of locally optimal trajectories for each $\theta$ via TrajOpt [53]. The optimal trajectory for a given $\theta$ is then defined as

$$\xi(\theta) := \underset{\xi \in \hat{\mathcal{T}}}{\arg\min} \ \theta^T \phi(\xi)$$

**Proxy Reward.** For this case study, the robot begins by asking the human designer for a proxy reward (cost) function. It is difficult for humans to provide proxies that work across all environments [2], so the robot asks for a proxy that produces the desired behavior in the two training environments. The human can provide the proxy weights: $[0.55, 0.55, 0.55]$ and produce trajectories that match those of $\xi_{\theta^*}$ (Figure 2 depicted in orange). Providing a proxy applies constraints that shrink our feasible set from $\Theta$ to $\mathcal{F}_{\text{proxy}}$:

$$\mathcal{F}_{\text{proxy}} = \{\tilde{\theta} : \theta^{*T}\phi(\xi_{\tilde{\theta}}^{(1)}) \geq \theta^{*T}\phi(\xi_\theta^{(1)}), \ \theta^{*T}\phi(\xi_{\tilde{\theta}}^{(2)}) \geq \theta^{*T}\phi(\xi_\theta^{(2)}) \quad \forall \, \theta \in \Theta\},$$

where $\xi_\theta^{(i)}$ denotes the optimal trajectory[3] w.r.t. cost parameter $\theta$ in environment $i$. The new feasible set $\mathcal{F}_{\text{proxy}}$ contains only the parameters $\theta$ that produce optimal trajectories with respect to the true weights $\theta^*$ in environments 1 and 2. Although it is a subset of the original feasible set $\Theta$, the new feasible set $\mathcal{F}_{\text{proxy}}$ is still a reasonably large set (Figure 2, top, middle, orange area). Furthermore, although the proxy produces optimal trajectories in environments 1 and 2, it does not necessarily for environments 3 and 4. Figure 2 (top, right) illustrates the different trajectories that result from optimizing different $\theta \in \mathcal{F}_{\text{proxy}}$. To further narrow our feasible set, we will ask for another form of feedback: Improvement.

**Improvement.** The robot will now (actively) provide a nominal trajectory, and ask the human to improve it, i.e. alter the trajectory to better suit their preferences. Suppose the robot presents the human with the nominal trajectory shown in gray (Figure 2, middle, left). This nominal trajectory is inefficient, staying too close to the table. Based on $\theta^*$, the human could provide the improved orange trajectory (Figure 2, middle, left) that is more efficient and doesn't emphasize closeness to the table as much. This improvement reduces our feasible set from $\mathcal{F}_{\text{proxy}}$ to $\mathcal{F}_{\text{improvement}}$:

$$\mathcal{F}_{\text{improvement}} = \{\theta : \theta^T\phi(\xi_{\text{improved}}) \geq \theta^T\phi(\xi_R) \quad \theta \in \mathcal{F}_{\text{proxy}}\}.$$

Figure 2 (middle, middle) shows the effect of applying this constraint, shrinking the orange feasible set. The feasible set has shrunk, but not enough to guarantee optimal behavior in all environments.

Figure 3: Environments used for experiments on active selection of feedback. (Top) These four environments were used during "training". (Bottom) These four environments were held as a test set to measure maximum and average regret.

The improvement establishes that closeness to the table should not come at the cost of efficiency. As a result, it removes the red trajectory in environment 3, which greatly traded off efficiency for proximity to the table (Figure 2, middle, right). To further fine tune, we will ask the human to answer a trajectory comparison.

**Trajectory Comparison.** The robot presents the human with two trajectories (Figure 2 bottom, left, orange and gray) and asks which incurs less cost. The human answers "orange", the trajectory that prioritizes efficiency over distance to the table. This comparison feedback shrinks our feasible set from $\mathcal{F}_{\text{improvement}}$ to $\mathcal{F}_{\text{comparison}}$:

$$\mathcal{F}_{\text{comparison}} = \{\theta \in \mathcal{F}_{\text{improvement}} : \theta^T \phi(\xi_{\text{orange}}) \geq \theta^T \phi(\xi_{\text{gray}})\} .$$

We finally see a very small orange feasible set (Figure 2, bottom, middle). Appropriately, in all four environments now, every $\theta \in \mathcal{F}_{\text{comparison}}$ produces a trajectory $\xi_\theta$ s.t. $\phi(\xi_\theta) = \phi(\xi_{\theta^*})$. This is illustrated in Figure 2 (bottom, right) as only the optimal green trajectory remains in each environment.

Our case study showcases the usefulness of combining types of feedback. A designer might start with their best guess at a reward function, the robot might misbehave in new environments, the designer or even end-user might observe this and intervene to correct or stop the robot, etc. – over time, the robot should narrow in on what people actually want it to do.

## C   Actively selecting which type of feedback to use

Given we can mix and match types of feedback, we may also wonder what is the best *type* to ask for at each point in time. The probabilistic model defined by reward-rational choice hints at how to select the feedback type – pick the one that maximizes expected information gain. We point this out, not because using information gain as an active learning metric is a new idea, but because the ability to use it arises immediately as an application of the formalism.

Suppose we can select between $n$ types of feedback with choice sets $\mathcal{C}_1, \ldots, \mathcal{C}_n$ to ask the user for. Let $b_t$ be the robot's belief distribution over rewards at time $t$. The type of feedback $i^*$ that (greedily) maximizes information gain for the next time step is

$$i^* = \underset{i \in [n]}{\arg\max}\, I(r_t; c_i^*) = \mathbb{E}_{r_t, c_i^*} \left[ \log \left( \frac{p(c_i^* \mid r_t)}{\int_{r_t \in \mathcal{R}} p(c_i^* \mid r_t) b_t(r_t)} \right) \right],$$

where $r_t \sim B_t$ is distributed according to the robot's current belief, $c_i^* \in \mathcal{C}_i$ is the random variable corresponding to the user's choice within feedback type $i$, and $p(c_i^* \mid r_t)$ is defined according to the human model in Equation 1.

Figure 4: Statistics computed over 10 iterations of our greedy maximum information gain algorithm. We notice that demonstrations (purple) are initially very information dense but quickly flatten out, whereas comparisons (cyan) obtain more information but less efficiently. We notice that combining the two methods (orange) inherits the positive aspects of both, the efficiency of demonstrations with the precision of comparisons.

To showcase the benefit of actively selecting feedback types, we run an experiment with demonstrations and comparisons. We measure regret (maximum and expected difference, on holdout environments, in ground truth reward between 1) optimizing with ground truth vs. 2) optimizing with the learned reward). We manipulate whether we have access to demonstrations only, comparisons only, or both, as well as the number of feedback instances queried.

One may initially wonder whether comparisons are necessary, given that demonstrations seem to provide so much information early on. Overall, we observe that demonstrations are optimal early on, when little is known about the reward, while comparisons become optimal later, as a way to fine-tune the reward (Fig. 4 shows our results). The observation also serves to validate the approach contributed by [49, 25] in the applications of motion planning and Atari game-playing, respectively. Both papers manually define the mixing procedure we found to be optimal: initially train the reward model using human demonstrations, and then fine-tune with comparisons.

**Experiment Details** We tested 3 different active learning methods: active querying of demonstrations, active querying of comparisons, and active querying of demonstrations and comparisons, across 8 different gridworld environments depicted in Figure 3. The top 4 environments were used in training while the bottom 4 were held for testing. Each environment $e$ is a 25x25 gridworld MDP with a linear

reward function in 3 features: RGB color values of each pixel. We assign each $e$ with 10 different start goal pairs $(s, g)$ from which the algorithms can ask queries. The goal of each algorithm is to efficiently recover a ground truth reward $r^*$ through querying.

Since our rewards are linear in RGB, the feasible reward set $\mathcal{R}$ consists of 3D parameters that weight the value of each feature in the reward function. $\mathcal{R}$ can be constrained to the surface of the 3D unit sphere since reward functions in MDPs are scale invariant. We uniformly discretize points at the surface of the 3D sphere to approximate $\mathcal{R}$ via $\hat{\mathcal{R}}$. To approximate $\Xi$, we first compute the optimal trajectory under each $r \in \hat{\mathcal{R}}$ to make $\{\arg\max_\xi \ r(\xi); \ r \in \hat{\mathcal{R}}\}$. We include trajectories that are not the result of optimizing reward functions by inserting noise into the value function when computing optimal trajectories as above.

**Demonstrations and Comparisons as Hard Constraints** The algorithms recover $r^*$ by narrowing a set of feasible rewards with active queries. We use $\mathcal{R}_i$ to denote the set of feasible rewards at iteration $i$ of querying. Demonstrations and comparisons shrink the feasible set in the following way:

$$\mathcal{R}_{i+1}^{\text{demo}}(\xi_d) = \{r : r(\xi_d) \geq r(\xi) \quad r \in \mathcal{R}_i; \quad \forall \xi \in \Xi\}$$

$$\mathcal{R}_{i+1}^{\text{comp}}(\xi_1, \xi_2) = \begin{cases} \mathcal{R}_{i+1}^{\text{comp}}(\underline{\xi_1}, \xi_2) = \{r : r(\xi_1) \geq r(\xi_2) \quad r \in \mathcal{R}_i\} & \xi_1 > \xi_2 \\ \mathcal{R}_{i+1}^{\text{comp}}(\xi_1, \underline{\xi_2}) = \{r : r(\xi_2) \geq r(\xi_1) \quad r \in \mathcal{R}_i\} & \xi_2 > \xi_1 \end{cases}$$

For our experiments, we performed the following greedy volume removal over possible $(s, g)$ pairs that we specified in each environment.

$$\mathcal{R}_{i+1} = \begin{cases} \mathcal{R}_{i+1}^{\text{demo}}(\xi_d^*) & V_{\text{demo}} < V_{\text{comp}} \\ \mathcal{R}_{i+1}^{\text{comp}}(\xi_1, \xi_2) & V_{\text{comp}} < V_{\text{demo}} \end{cases}$$

$$V_{\text{comp}} = \min_{(s,g)} \max \ \left\{\mathcal{R}_{i+1}^{\text{comp}}(\underline{\xi_1}, \xi_2), \ \mathcal{R}_{i+1}^{\text{comp}}(\xi_1, \underline{\xi_2})\right\}$$

$$V_{\text{demo}} = \min_{(s,g)} \mathbb{E}_{r^* \in \mathcal{R}_i}\left[|\mathcal{R}_{i+1}(\xi_{r^*}^{\text{demo}})|\right]$$

For demonstrations, we look for the $(s, g)$ pair that in expectation produces a demonstrations that leave the smallest feasible set (size of feasible set is volume or diameter described below). For comparisons, we look for the pair of trajectories $(\xi_1, \xi_2)$ that produce the minimum worst-case feasible region remaining. For the method with demonstrations and comparisons, we computed the above 2 metrics and select the feedback type with the smaller feasible region. We run this algorithm for 10 iterations and average our results across 50 different ground truth $r^*$. We plot several statistics for each iteration in Figure 4 including

$$\begin{cases} |\mathcal{R}_i| & \text{Volume at iteration i} \\ \sup_{r_1, r_2 \in \mathcal{R}_i} \|r_1 - r_2\|_2 & \text{Diameter at iteration i} \\ \max_{e;(s,g); \ r \in \mathcal{R}_i} \ r^*(\xi_{r^*}^{(e,s,g)}) - r^*(\xi_r^{(e,s,g)}) & \text{Max regret at iteration i} \\ \mathbb{E}_{e;(s,g); \ r \in \mathcal{R}_i}\left[r^*(\xi_{r^*}^{(e,s,g)}) - r^*(\xi_r^{(e,s,g)})\right] & \text{Avg regret at iteration i} \end{cases}$$

where $e$ is a holdout environment and $(s, g)$ is a start-goal pair in the MDP. Each metric is a proxy for how accurate our estimate of $r^*$ is. We notice that the combination of demonstrations and comparisons achieves lower volume, diameter, max regret, and average regret than demonstrations alone and that it achieves this in fewer iterations than comparisons alone.

# D   Meta-choice: a new source of information

In Section 4, we described a straight-forward way of combining feedback types: treat each individual feedback received as an independent reward rational choice, and update the robot's belief (Equation 6). However, the moment we open it up to multiple types of feedback, the person is not stuck with a single type and is actually choosing which type to use. *We propose that this itself is a reward-rational implicit choice, and therefore leaks information about the reward.* We call the choice of feedback "meta-choice", and in this section, we formalize it and empirically showcase its potential importance.

Figure 5: (Left) Environment with designated start (red circle), goal (green circle) and lava area (red tiles). The human can provide a correction (one of the green trajectories) or turn off the robot, forcing the robot to stop at the marked dot. (Middle) Belief distribution over rewards after the human provides feedback ($\beta_0 = 10.0$). Darker indicates higher probability. The metareasoning model is able to rule out more reward functions than the naive model. (Right) When the human's metareasoning has no signal ($\beta_0 = 0$), then the metareasoning (orange) and naive model (gray) perform equally well. As $\beta_0$ increases, the advantage of the metareasoning model also increases.

## D.1 Formalizing meta-choice

The assumption of conditional independence that the formulation in (6) uses is natural and makes sense in many settings. For example, during training time, we might control what feedback type we ask the human for. We might start by asking the human for demonstrations, but then move on to other types of feedback, like corrections or comparisons, to get more fine-grained information about the reward function. Since the human is only ever considering one type of feedback at a time, the conditional independence assumption makes sense.

But the assumption breaks when the human has access to multiple types of feedback at once because *the types of feedback the robot can interpret influence what the human does in the first place*.[4] If the human intervenes and turns the robot off, that means one thing if this were the only feedback type available, and a whole different thing if, say, corrections were available too. In the latter case, we have more information - we know that the user chose to turn the robot off rather than provide a correction.

Thus, our key insight is that the type of feedback itself leaks information about the reward, and the RRC framework gives us a recipe for formalizing this new source: we need to uncover the set of options the human is choosing from. The human has two stages of choice: the first is the choice between feedback types, i.e corrections, language, turn-off, etc. and the second is the choice within the chosen feedback type, i.e the specific correction that the human gave. Our formalism can leverage both sources of information by defining a *hierarchy* of reward-rational choice.

Suppose the user has access to $n$ types of feedback with associated choice sets $\mathcal{C}_1, \ldots, \mathcal{C}_n$, groundings $\psi_1, \ldots \psi_n$, and Boltzmann rationalities $\beta_1, \ldots, \beta_n$. For simplicity, we assume deterministic groundings. The set of choice sets $\mathscr{C}_0$ for the first-stage choice is $\{\mathcal{C}_1, \ldots, \mathcal{C}_n\}$ The grounding $\psi_0 : \mathcal{C} \to f_\Xi$ for the first stage choice maps a feedback type $\mathcal{C}_i$ to the distribution of trajectories defined by the human's behavior and grounding in the second stage:

$$\psi_0(\mathcal{C}_i) \; = \; \mathbb{P}(\xi \mid r, \mathcal{C}_i) = \sum_{c_i \in \mathcal{C}_i : \psi_i(c_i) = \xi} \mathbb{P}(c_i \mid r, \mathcal{C}_i) \,, \tag{9}$$

where, as usual, $\mathbb{P}(c_i \mid r, \mathcal{C}_i)$ is given by Equation 1. Instantiating Equation 1 to model the first-stage decision as well, results in the following model for the human picking feedback type $\mathcal{C}_i$:

$$\mathbb{P}(\mathcal{C}_i \mid r) = \frac{\exp\left(\beta_0 \cdot \mathbb{E}_{\xi \sim \psi_0(\mathcal{C}_i)}[r(\xi)]\right)}{\sum_{j \in [n]} \exp\left(\beta_0 \cdot \mathbb{E}_{\xi \sim \psi_0(\mathcal{C}_j)}[r(\xi)]\right)} \, , \qquad (10)$$

Finally, the probability that the human gives feedback $c^*$ is

$$\mathbb{P}(c^* \mid r) = \sum_i \mathbb{P}(c^* \mid r, \mathcal{C}_i) \cdot \mathbb{P}(\mathcal{C}_i \mid r) \qquad (11)$$

$$= \sum_i \left( \frac{\exp\left(\beta_i \cdot r(\psi_i(c^*))\right)}{\sum_{c \in \mathcal{C}_i} \exp\left(\beta_i \cdot r(\psi_i(c))\right)} \cdot \frac{\exp\left(\beta_0 \cdot \mathbb{E}_{\xi \sim \psi_0(\mathcal{C}_i)}[r(\xi)]\right)}{\sum_{j \in [n]} \exp\left(\beta_0 \cdot \mathbb{E}_{\xi \sim \psi_0(\mathcal{C}_j)}[r(\xi)]\right)} \right) . \qquad (12)$$

The first-stage decision can be interpreted as the human *metareasoning* over the best type of feedback. The benefit of modeling the hierarchy is that we can cleanly separate and consider noise at both the level of metareasoning ($\beta_0$) and the level of execution of feedback ($\beta_1, \ldots, \beta_n$). Noise at the metareasoning level models the human's imperfection in picking the optimal type of feedback. Noise at the execution level might model the fact that the human has difficulty in physically correcting a heavy and unintuitive robot.[5]

### D.2 Comparing the literal interpretation to meta-choice

We showcase the potential importance of accounting for the meta-choice in an experiment in a gridworld setting, in which an agent navigates to a goal state while avoiding lava (Figure 5, left). The reward function is a linear combination of 2 features that encode the goal and lava. The human has access to two channels of feedback: "off" and corrections. We simulate the human feedback as choosing between *feedback types* according to Equation 12. We manipulate three factors: 1) whether the robot is *naive*, i.e. only accounts for the information *within* the feedback type, or *metareasons*, i.e. accounts for the other feedback types that were available but not chosen; 2) the meta-rationality parameter $\beta_0$ modeling human imperfection in selecting the optimal type of feedback; and 3) the location of the lava, so that the rational meta-choice changes from off to corrections. We measure regret over holdout environments.

Figure 5 (left) depicts the possible grounded trajectories for corrections and for off. For the top, off is optimal because all corrections go through lava. For the bottom, the rational meta-choice is to correct. In both cases, we find that meta-reasoning gains the learner more information, as seen in the belief (center). For the top, where the person turns it off, the robot can be more confident that lava is bad. For the bottom, the fact that the person had the off option and did not use it informs the robot about the importance of reaching the goal. This translates into lower regret (right), especially as $\beta_0$ increases and there is more signal in the feedback type choice.

### D.3 What happens when metarationality is misspecified?

In our main metareasoning experiments, we assumed that the simulated human metareasoned with $\beta_0$ and that our algorithm somehow knew this quantity. However, in practice, we will not have access to $\beta_0$. This brings about an interesting question: What are the effects of inference under a misspecified $\beta_0$. What are the effects of overestimating or underestimating the human's rationality?

To test this, we designed an experiment in which our simulated human provided supervision with a fixed ground truth $r^*$ and $\beta_0^*$ while our algorithm performs belief updates with various $\beta_0$ above and below $\beta_0^*$. The first way to measure the extent of misspecification is to measure the KL divergence between the belief induced by $\beta_0^*$ and that induced by $\beta_0$.

$$D_{KL}(P(r \mid c^*) \| Q(r \mid c^*))$$

$$P(r \mid c^*) \propto P(c^* \mid r, \beta_0^*) \cdot P(r)$$

Figure 6: In each plot, the human operates under a true metarationality denoted by $\beta_0^*$. We measure performance drop from misspecification by computing the KL divergence and expected regret of the belief distribution over rewards for robots with misspecified metarationalities $\beta_0 \in [0.0, 10.0]$. (Top) The plots display the KL divergence between the true belief with $\beta_0^*$ and various misspecified beliefs. We notice that assuming metareasoning when the human does not metareason (left $\beta_0^* = 0$) results in significant divergence in the belief distribution. (Bottom) The plots show the expected regret for robots that learn, with misspecified $\beta_0$'s, from a human who gives feedback with $\beta_0^*$. As with *KL divergence*, the *expected regret* incurred by assuming metareasoning when the human does not metareason is high. Additionally, we note that a robot learning with $\beta_0 = \beta_0^*$ does not necessarily incur minimum expected regret (as mentioned below in the text).

$$Q(r \mid c^*) \propto P(c^* \mid r, \beta_0) \cdot Q(r)$$

Additionally, we wanted to measure the expected regret given a human that provides supervision with rationality $\beta_0^*$ and the algorithm that performs belief updates with rationality $\beta_0$.

$$\mathbb{E}[\text{Regret} \mid c^*, C_i, r^*, \beta_0] = \sum_{r \in \mathcal{R}} (r^*(\phi(r^*)) - r^*(\phi(r)))$$
$$\cdot \mathbb{P}(r \mid c^*, C_i, \beta_0)$$

$$\mathbb{E}[\text{Regret} \mid \beta_0] = \sum_{r^* \in \mathcal{R}} \sum_{i \in \mathcal{C}_0} \sum_{c^* \in C_i} \mathbb{E}[\text{Regret} \mid c^*, C_i, r^*, \beta_0]$$
$$\cdot \mathbb{P}(C_i \mid r^*, \beta_0) \cdot \mathbb{P}(c^* \mid C_i, r^*, \beta_0)$$

We plot the results in Figure 6 averaged over 50 randomly sampled reward functions and $beta_0 \in [0.0, 10.0]$. We notice that when the human does not metareason ($\beta_0^* = 0.0$, the KL divergence in the belief distribution update is large. In comparison, with any moderate level of metareasoning $\beta_0^* = 2.5, 5.0, 7.5$, the KL divergence is very low. We notice this too in the expected regret. Note that the minimum expected regret is not achieved by $\beta_0 = \beta_0^*$. This is because $\beta_0^*$ is used to compute the frequency at which the human provides each type of feedback as an answer. Simply matching $\beta_0$ with $\beta_0^*$ doesn't guarantee minimum expected regret (the optimal $\beta_0$ for minimizing expected regret is a function of $\beta_0^*$). These experiments suggest that if we detect that the human is poor at metareasoning (low $\beta_0^*$), it is safer to drop the metareasoning assumption. However, if the human is displaying metareasoning, we can leverage this to improve learning.

## Footnotes

[3]In our case study, the optimal trajectory is unique.

[4]We note that this adaptation by the human only applies to types of behavior that the human uses to purposefully communicate with the robot, as opposed to sources of information like initial state.

[5]Although we modeled rationality with respect to the reward $r$ that the robot should optimize, we can easily extend our formalism to capture that the person might trade-off between that and their own effort – this is especially interesting at this meta-choice level, where one type of feedback might be much more difficult and thus people might want to avoid it unless it is particularly informative.