[Reviews · NeurIPS 2020]

Review 1

Summary and Contributions: This paper proposes a conceptual framework, reward-rational implicit choice, to unify many types of human feedback that can be incorporated in reward learning. Many types of feedback are unified under this framework, including common explicit sources of information (demonstrations, preferences) and less common implicit sources of information (turning off a robot, state of the world).

Strengths: The paper proposes a conceptually novel and elegant framework, reward-rational implicit choice. The authors provide a unified lens and analyze many instances of human feedback (or slightly modified versions) with the framework, which demonstrates that it is fairly universal for many existing and potentially other types of feedback. The presentation of the framework is very clear and enjoyable to read. With the recent exploration of many different forms of human feedback (both explicit and implicit), it is good to see a unifying perspective of learning from human preferences. Section 3 covers this extensively, and the included tables are very helpful. The toy example (Figure 1) was a great way to ground the various approaches, although the text in the axes is rather small, and it was only possible to understand the plots while reading the text. The idea of actively deciding which feedback type to ask for is quite interesting, and is a good line for future exploration. Apart from using the framework to query an appropriate type, it would also be interesting to conduct human experiments to see how well the framework reflects human choices of which type of feedback to provide, assuming the system can handle them.

Weaknesses: Although the inference procedure is conceptually clear, it would be helpful to work out the mathematical details, e.g., applied to the toy example Figure 1. Specifically, this means given the choice set in the first row and the grounding in the second row, computing the distribution over reward functions as given by Equation 3. It was unclear whether the example in Figure 1 was meant to be conceptual only, or whether the inference algorithm can be applied "as is" to produce the bottom row of Figure 1. This would provide greater confidence to the reader about how to apply the framework (without needing to read all the related papers), and may give further insight into the potential computational complexity of some of the methods, in particular how they compare with each other. Of course, there is no room in the paper to do this, but it would be a helpful supplementary section, and could improve the impact of the work because readers will gain deeper understanding. It would also be helpful to discuss the human rational choice model. Although there are many references to works in the behavioral sciences, as well as a theoretical justification in the appendix, it would be useful to know how good of a fit this is experimentally (in previous papers), since the entire inference framework rests on the assumption that Equation 1 is a good model. Additionally, does any previous work consider how sensitive this model may be to misspecified beta values? Finally, the author(s) suggest this formalism may not apply to all types of feedback -- are there known examples of this? If not, would it be possible to imagine certain feedback (natural or contrived) that does not fit well with the framework?

Correctness: The claims appear correct, and the derived entities in Tables 1 and 2 appear correct. There is no experimental section.

Clarity: The paper is very clear and well-written. It was a pleasure to read!

Relation to Prior Work: As a survey of previous work under a new framework, previous work is clearly discussed, and this work clearly differs in that it is a unifying perspective on many previous work.

Reproducibility: Yes

Additional Feedback: - There are two repeated sentences (lines 204-207 vs. lines 208-211). - The choice set for demonstration (row 1, col 2 of Figure 1) was unclear -- why is it not a complete grid? Is a demonstration that goes on the rug but not in a diagonal path possible? - The grounding of correction (row 2, col 3 of Figure 1) was unclear -- why does the correction cause the grounding to be much further away? From the description, it would make sense for the rest of the trajectory to move slightly to the upper left, but the change shown in the figure is quite dramatic. - The grounding of reward and punishment (row 2, col 8 of Figure 1) was also unclear -- what do the gray and orange trajectories indicate, and why are they the grounding of +1 and -1 respectively?


Review 2

Summary and Contributions: The authors survey and provide a unifying formalism for learning from human choices/feedback/demonstrations and more. They comb the literature thoroughly and describe a whole suite of techniques which are broken down into sets of options and a grounding function which maps the instruction to a set of trajectories.

Strengths: The paper is very well written, scholarly and well researched. The cited works are thorough. I found the formalism to be intuitive and can imagine myself using this as a resource to cite both the diversity of feedback mechanisms available and to use this formalism to develop new ones. This topic is definitely of interest to the NeurIPS community.

Weaknesses: This work falls short of the best kinds of formalisms operate as a kind of periodic table. By laying things out in the right way, a new method is revealed. This work does not reveal a new method or new way of choosing between methods. The authors propose some future directions of how these techniques could be applied in the discussion and I think a more mature version of this manuscript would actually develop or test those techniques rather than leaving them to future work. By testing these methods the novelty of this work be ensured.

Correctness: Yes

Clarity: Yes

Relation to Prior Work: Yes

Reproducibility: Yes

Additional Feedback: Thank you to the authors for their response. The new experiments and analyses address my points and I now recommend the paper be accepted. In the final version of the manuscript, the authors should definitely address the cost of feedback as suggested by the other reviewer. I think this would also help highlight how this framework can be used as a decision making tool to decide which kind of feedback will be most most informative to the agent in terms of cost of feedback (which is likely to vary across situations in many ways).


Review 3

Summary and Contributions: This paper presents a general framework for reward learning from human behavior or feedback. The authors propose ‘reward-rational (implicit) choice’ as the formalism for unifying all variants of reward learning schemes in the literature. This paper suggests that many works in the literature of reward learning can be viewed as reasoning over a set of choices and defining a grounding function for interpreting a particular human behavior or feedback as robot behaviors. The proposed formalism gives new insights into prior works, showing how reward learning unifies many proposed learning from human problems including some in the reinforcement learning regime, and also inspires explorations of new sources of information, combination of different feedback types and the idea of meta-active learning on the type of feedback to use.

Strengths: This paper provides a theoretical tool for analyzing existing algorithms for reward learning. The paper derives the corresponding instantiation of the proposed framework for selected prior works with a detailed explanation of how the derivation is equivalent to the original methods. The illustrative example problem and the corresponding visualization is nicely used for comparing different schemes of existing methods and feedback types. The comprehensive survey of prior works with the new formalism is well written with thoughtful discussions. The proposed unifying formalism is conceptually novel and inspiring, which will significantly influence the research fields of robot learning, computational human modeling, and human-robot interaction.

Weaknesses: One limitation of this framework is that there is no notion of cost associated with different types of feedback. When evaluating all different kinds of human feedback practically, it is often important to compare each of their advantages and disadvantages given different assumptions. For example, different types of feedback probably require different levels of human (cognitive) effort and different tasks would also induce different levels of human (physical) effort to demonstrate/intervene (for humans, some tasks such as driving are easier to demonstrate while others such as flying a helicopter are easier to judge). In the proposed framework, there is no notion of cost associated with providing feedback and therefore the proposed meta-active learning method for choosing feedback types only depends on information gain in the reward space but not reasoning over potential communicative effort required. ------------------------------------- Edit: In the rebuttal, the authors promise to discuss the above concern in the revised paper. Since the authors also pointed out in the rebuttal that the active learning algorithm based on the meta-choice model will be an important contribution of this work, it is important to identify how cost can be incorporated in such an algorithm.

Correctness: Yes. The derivation of prior works to instantiations of the proposed framework is correct and the gridworld example demonstrates how different feedback types would work within the same problem with the same formalism.

Clarity: Yes, this paper is well organized logically and the conceptual ideas are very well explained with detailed examples.

Relation to Prior Work: Yes. This paper survey prior works with the proposed framework. -------- Edit: It is also important to notice that the proposed framework inspires new algorithms and therefore is not just a survey of prior works.

Reproducibility: Yes

Additional Feedback: Considering other reviews and the author's feedback, there seems to be a concern on the level of novelty of this paper's contribution. With my limited experience in this field, I believe the contribution of this paper is still significant through providing a unifying framework for reward learning and inspiring new algorithms that can leverage the human's meta-choice as additional information. However, I moved my overall score for this paper from the initial assessment of 8 to 7 in light of other reviewers' comments. With limited experience reviewing for NeurIPS, I also moved my confidence score from 4 to 3.


Review 4

Summary and Contributions: This paper presents a framework that unifies many types of inputs for reward learning and provides the mathematics to convert all types of input into a consistent form. The framework is demonstrated through a hypothetical case study.

Strengths: soundness: The framework is theoretically grounded and this reviewer could not find any major issues with the mathematical foundations used. Significance: The paper as a call for more people to consider "multi-modal" (multi-input-type might be more correct) reward learning is one reason why this paper might be seen as notable. Relevance to NeurIPS: reward learning is an important topic in the NeurIPS community.

Weaknesses: Evaluation: There is no empirical evaluation. Arguably there could not be any empirical evaluation of a framework, since a framework is meant to help us categorize and understand a phenomenon. However, it might have been possible to show that a simple agent could be trained with each type of reward input covered by the framework. It may be that the learning is very inefficient. Significance of the work is unclear. What are we able to do or think about that we were not able to do or think about prior to the framework? While it makes intuitive sense that each type of input to reward learning can be transformed into each other and into a common form, many of those transformations are inefficient (e.g., consider all trajectories that are consistent with a language utterance). The fact that we didn't start with a unification may be because we require more efficient representations and formulations that each.

Correctness: The mathematical formulations appear mostly correct. In Corrections, the equation C = Q - Q does not seem like it can be right, but may just be a typo. As noted above, there is no empirical methodology as it is difficult to prove the validity of a framework, which is a tool for thinking about a problem. However, in this case, one could convert a number of input types into the unifying form and prove that an agent can learn. This will validate the mathematical formulae. There is another way in which RL agents learn from "off" buttons. The work on Big Red Button Problems (Google; Riedl) indicates that agents can learn from the fact that more reward is procured in longer traces than traces that are aborted early with the "off" button. Why does the grounding of language have to result in a uniform distribution over consistent trajectories? Language could be of the form "prefer to avoid water", which would imply a weighting of trajectories. Also, learning to use commonsense in reward learning (Harrison & Riedl) would imply a soft filtering and thus a non-uniform distribution over trajectories. Not mentioned in the paper are learning from critique and learning from advice (See Andrea Thomaz, Karen Feigh), who have also conducted experiments in how humans prefer to teach agents.

Clarity: The paper is mostly clear. In table 1, language and initial state, it is not immediately clear that unif() is the uniform distribution. The rug example in Fig 1 needs to be introduced more completely. It is very hard to follow what is going on in Fig 1 and some discussion of the example domain may help. The bottom row of fig 1, feasible rewards, is inscruitable; what does the sphere represent? Is this the cartesian space where reward is given?

Relation to Prior Work: The related work is extensive. The paper partially serves as a lit review.

Reproducibility: Yes

Additional Feedback:

[Author Response · NeurIPS 2020]

We thank the reviewers for their time and thoughtful feedback. They were kind to refer to the formalism as novel, elegant, and inspiring, and to point out that the instantiations demonstrated it is fairly universal for many existing and potentially new types of feedback. Even **R2**, our harshest critic, pointed that "I can imagine myself using this as a resource to cite both the diversity of feedback mechanisms available and to use this formalism to develop new ones." This is what we were hoping for! In what follows, we hope to alleviate **R2**'s main concern, and we take the opportunity to respond to other points the reviewers brought up.

**Usefulness.** **R2**'s main critique is that there isn't a new method falling out of the formalism. **R4** also asks "What are we able to do or think about that we were not able to do or think about prior to the framework?" As **R2** is actually aware, our discussion does point to several things, including the ability to combine and actively select the input types, but here we would like to emphasize the meta-choice. *The moment we said that there are multiple types of available feedback to a person, and that we should think the person is making an implicit choice within each type, it become clear that the type of feedback is itself an implicit choice* – our realization was that it too leaks information about the reward. We actually find this to be a really compelling example of exactly what **R2** and **R4** seem to be looking for! Now, **R2** does make a fair point that it'd be good to develop this further.

Unfortunately there is no way to squeeze this in the paper (and still explain the formalism properly). We also really don't want it to distract from the formalism as the main contribution, which **R2** acknowledged can already be useful in developing new types of feedback. But we have run some experiments with meta-choice, and will put these in the appendix. The experiments simulate a user choosing between corrections and "off" when a robot was dealing with "lava". By understanding this as a reward rational implicit choice, the robot is able to understand more

about the reward: on the bottom of the figure, if the human did a correction, and it knows the person had the "off" option and didn't use it, that tells it about the importance of reaching the goal. The plots in the center compare the belief ofer the weight on goal and lava for both naive and this "meta" inference, showing larger entropy reduction with the latter.

**Actual implementation.** **R1** and **R4** want to see the input types actually converted to the framework and implemented. We want to clarify that this is what is happening in Fig.1. Those are the actual reward inferences coming from each type, produced by our implementation (granted, in a simple domain, for illustration purposes to see how the types compare). **R4** might be suggesting taking this "evaluation" a step further, i.e. an analysis where each feedback type is used repeatedly. We've done experiments where the agent actively chooses which feedback is most informative which we could add to the appendix. But we do want to (respectfully) ask the reviewer to consider that this is one of the *many* things that would be useful to do with this framework, which is what makes the framework such a meaningful contribution.

**Language.** **R4** asks why language has to result in a uniform distribution over trajectories. We apologize, it does not have to: the formalism, as seen in eq. 1, maps choices to distributions over trajectories. Whether the distribution is uniform or not depends on the language model. This was our mistake, we will clarify! Thank you for bringing this up.

**The rationality assumption.** **R1** rightfully asks whether people are actually Boltzmann-rational. While this assumption has nice properties derived in our appendix and seems to have been useful in the works instantiating this formalism, it is also wrong, at least when applied naively. Recent work has explored how maybe people who seem to be irrational are actually rational, but under different assumptions. For instance, they might assume a different dynamics model, a different observation model, or use a different planning horizon. But any such improvements in human modeling can then translate to the formalism, now that we have all types of work under one unifying umbrella. One useful thing to note is that an agent can potentially detect when this assumption is wrong by detecting that no reward function explains the human's choice sufficiently well.

**Cost.** **R3** rightfully points out that different feedback types have different costs. When doing active learning, the agent could trade off between information gain and user cost, or have a cost budget. We'll be sure to discuss this in the paper! We also note that different types might be associated with different rationality parameters, which naturally affect their informativeness.

[Meta-Review · NeurIPS 2020]

There were initial concerns from the reviewers about the contribution of this framework and whether it really changes what our fundamental understanding of the problem or practical capabilities, or if is more of a literature review with an interesting but not particularly impactful framework / unifying theory. After the author response and discussion, the reviewers came to the conclusion that this framing does, in fact, increase our understanding into how to compare many existing algorithms / feedback types, and that the meta choice model is a sufficiently novel and useful insight.